# Toxic and Phenotypic Effects of AAV_Cre Used to Transduce Mesencephalic Dopaminergic Neurons

**DOI:** 10.3390/ijms23169462

**Published:** 2022-08-21

**Authors:** Larissa Erben, Jacqueline P. Welday, Ricardo Murphy, Andres Buonanno

**Affiliations:** Section on Molecular Neurobiology, Eunice Kennedy Shriver National Institute of Child Health and Human Development, National Institutes of Health, Bethesda, MD 20892, USA

**Keywords:** toxicity, stereotaxic injection, Cre recombinase, dopamine, ventral tegmental area, adeno-associated virus, tyrosine hydroxylase

## Abstract

A popular approach to spatiotemporally target genes using the *loxP*/Cre recombination system is stereotaxic microinjection of adeno-associated virus (AAV) expressing Cre recombinase (AAV_Cre) in specific neuronal structures. Here, we report that AAV_Cre microinjection in the ventral tegmental area (VTA) of ErbB4 Cyt-1-floxed (ErbB4 Cyt-1^fl/fl^) mice at titers commonly used in the literature (~10^12^–10^13^ GC/mL) can have neurotoxic effects on dopaminergic neurons and elicit behavioral abnormalities. However, these effects of AAV_Cre microinjection are independent of ErbB4 Cyt-1 recombination because they are also observed in microinjected wild-type (WT) controls. Mice microinjected with AAV_Cre (10^12^–10^13^ GC/mL) exhibit reductions of tyrosine hydroxylase (TH) and dopamine transporter (DAT) expression, loss of dopaminergic neurons, and they behaviorally become hyperactive, fail to habituate in the open field and exhibit sensorimotor gating deficits compared to controls microinjected with AAV_GFP. Importantly, these AAV_Cre non-specific effects are: (1) independent of serotype, (2) occur with vectors expressing either Cre or Cre-GFP fusion protein and (3) preventable by reducing viral titers by 1000-fold (10^10^ GC/mL), which retains sufficient recombination activity to target floxed genes. Our studies emphasize the importance of including AAV_Cre-injected WT controls in experiments because recombination-independent effects on gene expression, neurotoxicity and behaviors could be erroneously attributed to consequences of gene ablation.

## 1. Introduction

The neuregulins (NRG1-4) are a family of neuronal factors harboring an EGF-like domain that bind and activate the ErbB family (ErbB1–ErbB4) of receptor tyrosine kinases [1,2]. In neural tissue, ErbB2 and ErbB3 are mostly expressed in glia, whereas ErbB4 is the major neuronal receptor. Importantly, genetic variants of NRG1–NRG3 and ErbB4 are associated with an increased risk for schizophrenia and its endophenotypes [2,3,4,5,6]. ErbB4 is expressed in most dopaminergic neurons (92–99%) of the substantia nigra pars compacta (SNc) and the ventral tegmental area (VTA) [7,8,9], and NRG/ErbB4 signaling regulates dopamine homeostasis and cognitive functions [9,10,11,12,13,14]. Mice that lack ErbB4 specifically in dopaminergic neurons (TH-Cre; ErbB4^fl/fl^) exhibit an imbalance of basal extracellular dopamine levels in distinct projection areas, including the medial prefrontal cortex (mPFC), which coincides with cognitive deficits in spatial and working memory as well as deficits in the acquisition of tasks [9].

Four different ErbB4 isoforms are generated by alternative splicing of two exons encoding the extracellular juxtamembrane domain (JMa or JMb), and by the inclusion or exclusion of a 48bp exon in the cytoplasmic domain (Cyt) to generate ErbB4 Cyt-1 or Cyt-2 isoforms, respectively. This exon renders ErbB4 Cyt-1 isoforms the capacity to activate phosphatidyl-inositol-3-kinase (PI3K)/Akt downstream signaling [15]. Importantly, ErbB4 Cyt-1 receptor signaling and expression levels are altered in the dorsal lateral prefrontal cortex of persons with schizophrenia [4,16,17,18,19]. In order to specifically investigate the functional role of ErbB4 Cyt-1 receptors in dopaminergic neurons, we began by generating mice harboring a floxed allele of ErbB4 mice (ErbB4 Cyt-1^fl/fl^; described elsewhere by [20]). In the process of characterizing ErbB4 Cyt-1^fl/fl^ mice stereotaxically microinjected with AAV_Cre in the VTA, we unexpectedly found that AAV_Cre injections at commonly used titers (10^12^–10^13^ GC/mL) affected the behaviors and phenotypes of both ErbB4 Cyt-1^fl/fl^ and WT control mice similarly (see below).

Although use of the Cre/*loxP* system combined with stereotaxic microinjection of AAV_Cre has revolutionized studies on brain development and function [21,22,23,24,25], few studies have addressed the potential cytotoxicity of Cre recombinase expression mediated through AAVs in specific neuronal populations [26,27,28,29]. Notably, many of these studies were performed with concentrations that predated efficient technological advances that now regularly yield titers with thousands-fold more genome copies. Numerous recent studies routinely inject these undiluted AAV_Cre preparations (10^13^ GC/mL) into the brain without analyzing their effects on specific anatomical or neuronal populations or on behaviors of WT control mice.

Because of the aforementioned unspecific effects that we observed of AAV_Cre on the behaviors of ErbB4 Cyt-1^fl/fl^ mice, in this study we analyzed the effects of different AAV serotypes and AAV_Cre titers stereotaxically microinjected into the VTA of adult ErbB4 Cyt-1^fl/fl^ and WT control mice. We found that AAV_Cre microinjections (10^13^ GC/mL) cause a severe downregulation of tyrosine hydroxylase (TH) and dopamine transporter (DAT) expression, key proteins that regulate dopamine homeostasis in brain and result in hyperactivity, reduced habituation and deficits in sensorimotor function. Importantly, here we demonstrate that the effects of AAV_Cre are independent of viral serotype. Moreover, Cre-dependent toxicity can be circumvented by reducing the amount of viral injection by approximately 1000-fold (10^10^ GC/mL), while still effectively recombining flox-targeted genes. The results of this study underscore the need of appropriate controls when studying gene ablation by viral-mediated Cre recombinase expression that include testing the effects of AAV_Cre (not only AAV or AAV_GFP) in wild-type mice; otherwise, it is impossible to discriminate between non-specific (e.g., cytotoxic) changes by Cre recombinase and phenotypes caused by the specific recombination of the target *loxP* sites.

## 2. Results

### 2.1. AAV_Cre Microinjections into the VTA Provoke Hyperactivity

In order to assess the effects of selectively ablating the ErbB4 Cyt-1 receptor isoform in adult dopamine neurons, we bilaterally injected into the ventral tegmental area (VTA) of Cyt-1^fl/fl^ mice GFP-tagged AAV1_Cre (AAV1_Cre-GFP) driven by the synapsin promoter (refer to Section 4.2 for details). As is commonly used for non-targeting controls, Cyt-1^fl/fl^ mice injected with AAV1_GFP (i.e., virus lacking Cre recombinase) served as negative controls (Figure 1A; for details see Appendix A). We analyzed a battery of behaviors previously reported to be altered in ErbB4 null mice or dopaminergic neuron-targeted ErbB4 mice, such as locomotor activity, sensorimotor gating, and Y-maze performance [9,30,31]. As shown in Figure 1B,C, mice microinjected with AAV1_Cre-GFP became severely hyperactive and were unable to habituate to the new environment compared to AAV1_GFP control-injected mice. Anxiety in the elevated plus maze (Figure 1D) and spontaneous alternation in the Y-maze (Figure 1E) were unchanged in either AAV1_Cre-GFP or AAV1_GFP injected Cyt-1^fl/fl^ mice. However, AAV1_Cre-GFP injections into the VTA resulted in sensorimotor gating deficits assessed in the prepulse inhibition task (Figure 1F). The increased locomotor activity and reduced prepulse inhibition observed in VTA-targeted AAV1_Cre-GFP ErbB4 Cyt1^fl/fl^ mice would have been consistent with prior studies using ErbB4 null mice, if it were not for the fact that in a different project in our laboratory, using mice harboring a constitutive Cyt-1 mutation generated by somatic deletion of the Cyt-1 exon (referred to as ErbB4 Cyt-1 KO mice; see Methods), we did not observe these phenotypes [20].

In order to confirm the unexpected phenotypes we observed in Cyt-1^fl/fl^ mice injected in the VTA with AAV1_Cre-GFP, and because the Cre-GFP fusion protein could potentially cause transcriptional shut-down due to agglomerations in the nucleus [29], we injected a second cohort of Cyt-1^fl/fl^ mice with a different batch and serotype of non-tagged AAV9_Cre mixed with AAV9_DIO-GFP (hereafter simply referred to as AAV9_Cre) under the control of synapsin promoter (Figure 2A; for details see Appendix A); mice injected with AAV9_GFP were again used as negative controls. Similarly to AAV1_Cre-GFP-injected mice, AAV9_Cre-injected mice were hyperactive in a novel environment and spent less time in the center of the open field compared to AAV9_GFP control mice (Figure 2B,C). AAV9_Cre-injected mice also did not show deficits in the elevated plus maze (Figure 2D) or spontaneous alternation in the Y-maze (Figure 2E). In contrast to AAV1_Cre-injected Cyt-1^fl/fl^ mice (Figure 1F), AAV9_Cre injections did not elicit deficits in sensorimotor gating (Figure 2F). Thus, VTA microinjections of ErbB4 Cyt-1^fl/fl^ mice with either AAV1_Cre or AAV9_Cre, which selectively target the Exon 26 Cyt-1 exon in the mesencephalon, caused hyperactivity in these mice.

### 2.2. AAV_Cre Injections in the Mesencephalon Can Cause Loss of Key Protein Expression and Lesions

These findings prompted us to closely analyze the microinjection sites and status of the mesencephalic dopaminergic neurons after completion of the behavioral analyses. As shown in Figure 3, microinjections of AAV1_Cre-GFP and AAV9_Cre, as well as the negative control injections (AAV1_GFP and AAV9_GFP, Figure 3A,E), resulted in a widespread transduction in the midbrain assessed by analyzing relative GFP expression levels (Figure 3B,F). Of note, AAV1_Cre-GFP was visualized by nuclear GFP (Figure 3B, lower panel) whereas in all other cases GFP expression is somatic. Unexpectedly, we observed that cells immunoreactive for TH and DAT, markers used to identify dopaminergic neurons, were dramatically reduced four weeks post-op in the VTA of mice injected with AAV1_Cre-GFP relative to controls injected with the AAV1_GFP virus that lacks Cre-recombinase (Figure 3B–D). While AAV9_Cre injections did not appear to alter DAT or TH protein levels four weeks post-injection (Figure 3F–H), all Cre-injected mice exhibited changes in the DAPI stain. DAPI-negative holes surrounded by a dense layer of nuclei appeared throughout the transduced area. These ring-like lesions have previously been described as “apoptotic rings” (Figure 3F’; [29]). Inconsistency in the loss of dopamine markers and appearance of lesions in mice transduced with Cre-expressing virus raised the concern that the phenotypes we observed were independent of ErbB4 Cyt-1 loss.

### 2.3. AAV1_Cre Injections in Wild-Type Mice Confirm That Loss of Protein Expression and Hyperactivity Are Unspecific 

Based on these observations, as well as a report published when this work was in progress [29], we investigated the effects of AAV1_Cre injections into wild-type (WT) mice to determine if the reductions of TH and DAT were related to the loss of ErbB4 Cyt-1 or non-specific effects of AAV1_Cre injection. For these experiments, we performed unilateral microinjections as an intra-animal control (Figure 4A). As observed in ErbB4 Cyt-1^fl/fl^ mice, microinjection of AAV1_Cre-GFP into WT mice also resulted in hyperactivity and loss of habituation to novelty when compared to AAV1_GFP control injections (Figure 4B,C). Importantly, immunostainings revealed a dramatic loss of TH and DAT expression in the ipsilateral (injected) side, but not in the contralateral (non-injected) side in the midbrain of AAV1_Cre-GFP-injected WT mice (Figure 4D–F). These experiments indicate that microinjection of AAV1_Cre-GFP at viral concentrations (~10^13^ GC/mL) regularly used in other studies [32,33] causes midbrain loss of TH and DAT, as well as induces behavioral phenotypes (i.e., hyperactivity).

### 2.4. Lowering AAV_Cre Titer Prevents Toxicity While Retaining Cre/loxP Recombination Activity

High levels of Cre recombinase expression driven by strong promoters in transgenic mice or by viral transduction have been associated with cell death and unspecific side effects [26,29,34]. To determine if decreasing Cre recombinase levels could reduce toxicity and behavioral alterations described earlier, we tested 10-fold (Appendix A) and 1000-fold (Figure 5) dilutions of the AAV9-Cre stock (~10^13^ GC/mL) in WT mice. Although at a 10-fold dilution we continued to observe increased activity in the open field (Appendix A), we found that WT C57BL/6J mice microinjected in the VTA with the 1000-fold diluted AAV9_Cre/AAV9_DIO-GFP mix (Figure 5A) did not exhibit behavioral changes in the open field and elevated plus maze (Figure 5B–D), and only mild effects on the Y-maze and at the 66 dB, but not 70, 74 and 78 dB, PPI response (Figure 5E,F). Importantly, as shown in Figure 5G, using AAV9_Cre at this lower titer was sufficient to effectively recombine the co-injected AAV9_DIO-GFP. Consistent with these observations, immunohistochemical analyses performed with tissues collected four weeks post injection did not reveal lesions or reductions in DAT or TH expression (Figure 5G–I). Taken together, these findings indicate that the use of AAV9_Cre at a titer of 1.0 × 10^10^ GC/mL is sufficient to drive Cre/*loxP*-mediated recombination in the VTA, yet avoid the AAV-induced neurotoxicity observed in this brain region when utilizing the more common titer of ~10^13^ GC/mL.

## 3. Discussion

Here, we provide evidence that microinjection of AAV-Cre into the VTA of WT mice at concentrations regularly used for gene-targeting studies (~10^13^ GC/mL) can cause severe neurodegenerative effects and behavioral alterations that are independent of Cre/*loxP* gene recombination. Namely, dopaminergic neurons reduce expression of key proteins involved in dopamine homeostasis (DAT and TH) after AAV1_Cre injections at ~10^13^ GC/mL, and mice injected with these viral titers become hyperactive, do not habituate in the open field and exhibit sensorimotor gating deficits. Importantly, we show that lowering the viral titer of AAV9_Cre by approximately 1000-fold (~10^10^ GC/mL) circumvents the effects of the virus on behaviors—at least under the studied conditions (serotype, timeline, assays)—but Cre-recombinase activity is sufficiently retained for the effective recombination of *loxP* sites. This work underscores the neurotoxic effects that high titers of AAV_Cre recombinase regularly used by numerous laboratories to target genes in the VTA can have on monoaminergic neurons. Because most of those studies targeting floxed genes expressed in dopaminergic VTA and substantia nigra neurons used as controls AAV_GFP microinjections into floxed mice but did not test the effects of AAV_Cre in WT mice, in the future it would be advisable to include as controls microinjections of AAV_Cre in non-floxed mice to assess for potential Cre-related toxicity.

### 3.1. Toxicity Associated with High AAV_Cre Recombinase Expression Levels

The phenotypes of mice following AAV_Cre microinjection into the VTA described herein are similar to a recent study that reported neurotoxic effects of Cre recombinase [29]; however, it is important to emphasize that several parameters between the experiments conducted in the studies differ. Taken together, the results from both studies indicate that the neurotoxic effects observed after AAV_Cre microinjection in midbrain dopaminergic nuclei occur independently of injection site (SNc vs. VTA), AAV serotype, laboratory that prepared the AAVs, mouse strain and times following viral microinjection. The phenotypic changes of dopaminergic neurons and hyperactivity in mice were observed with three different AAV serotypes (AAV1, AAV2 and AAV9) purchased from different vendors (Addgene and Penn Vector Core), thus reducing the likelihood that serotypes or contaminants in the AAV preparations account for the toxicity. Our experiments also clarified that these undesired side effects are not caused by Cre-GFP fusion protein but that they also occur with unconjugated Cre recombinase, which was an unresolved issue in the earlier study [29]. Moreover, toxicity mediated by Cre recombinase was observed in experiments performed with C57BL/6J (this study) and C57BL/6N [29] mice, suggesting that AAV_Cre could affect other mouse strains and species. Lastly, the two studies were conducted using different timelines between AAV_Cre injection and performing behavioral studies (2–4 weeks; this study vs. 2–4 months [29]), indicating that the effects of AAV_Cre toxicity on behavior are stable and long-lasting.

### 3.2. Different AAV_Cre Viral Vectors Can Elicit Overlapping and Distinct Phenotypes

While the phenotypes elicited by distinct AAV_Cre expression vectors were generally similar, there were differences with regard to the severity of the behavioral and neuroanatomical alterations. For example, the loss of DAT and TH expression was most severe following microinjections of AAV1_Cre-GFP (this study), to a lesser extent after AAV2_Cre-GFP injection [29], and it was not observed following injection of AAV9_Cre (this study). Interestingly, at four weeks post-injection mesencephalic lesions were observed in all mice injected with AAV9_Cre and only in a few mice injected with AAV1_Cre-GFP. Of note, it remains to be seen what mechanism drives distinctions among AAV serotypes, and if these differences may be attributable to various transduction efficiencies—a question that is beyond the scope of the present study. Similarly, the extent of hyperactivity caused by AAV_Cre differed between the viral vectors tested. Injections of AAV1_Cre-GFP predominantly resulted in the loss of adaption to the novel environment (Figure 1B), whereas AAV9_Cre increased the locomotor activity at all time points (Figure 2B). Moreover, AAV2_Cre-GFP injections in the SNc provoked high thigmotaxis [29], whereas both viral vectors tested in this study only slightly decreased the time spent in the center of the maze (Figure 1C and Figure 2C). In summary, these findings suggest that AAV_Cre microinjection can elicit a range of neuroanatomical and behavioral phenotypes depending on the serotype/viral vector used. While our findings cannot exclude the possibility that other serotypes, promoters and viral constructs may not elicit toxic side effects, they underscore the importance that all AAV_Cre viruses used should be tested in WT animals to control for non-specific effects of Cre recombinase that are independent from ablation of targeted genes.

### 3.3. High Cre Recombinase Levels Can Cause Damage That Is Preventable by Reducing Expression

Non-canonical cryptic or pseudo *loxP* sites in the genome are frequent (1.2/megabase) and can recombine with high efficiency, especially at high Cre expression levels [35,36]. DNA damage due to double strand breaks and nicks is repaired by non-homologous end joining, which can result in chromosome rearrangements [35,37,38,39]. Cre-induced DNA damage has been proposed to induce programmed cell death either by p53-mediated apoptosis or by autophagy [29,40]. It also has been proposed that cytotoxicity and apoptosis arises through low, but persistent, DNA damage that can result in phenotypes that can extend 3–18 weeks [26,41,42]. On the relatively short time scale in this study (four weeks), we did not find any evidence that the massive reduction in DAT and TH expression was due to cell death. However, on a longer time scale, we observed dramatic reductions in nuclei staining (data not shown) that are consistent with those of Cre-mediated programmed cell death [29].

To our knowledge, this is the first study to directly demonstrate that reduction of viral Cre expression by decreasing the titer of AAV_Cre stereotaxic injections from ~10^12^–10^13^ GC/mL to ~10^10^ GC/mL can circumvent the undesired neurotoxic effects in midbrain dopaminergic neurons while retaining sufficient recombinase activity to target floxed-alleles. Toxic effects due to the high expression of Cre recombinase were observed at commonly used viral titers (~10^12^–10^13^ GC/mL). Optimizing the concentration of the virus to an amount approximately 1000-fold lower than regularly used prevented toxicity and behavioral alterations while retaining recombination of a co-injected AAV_DIO-GFP reporter construct (Figure 5E–H). Therefore, diluting AAV_Cre from typical concentrations of ~10^12^–10^13^ to lower concentrations that still effectively recombine to target flox genes may be a general approach to resolving toxicity phenotypes associated with microinjections. Similarly, other studies using inducible Cre mice for recombination reported that administration of lower tamoxifen doses circumvents cytotoxicity and off-target effects [35,37,38,39]. A question that merits further investigation in the future is if the lower AAV_Cre titers used here to transduce mesencephalic neurons will be applicable to target recombination of genes in different brain regions and neuronal subtypes.

In conclusion, AAV_Cre-mediated recombination has become a ubiquitously used tool in the field of neuroscience [25]. However, it will be important to carefully differentiate the phenotypes that result from the manipulation of targeted genes (e.g., gene deletion, optogenetic, chemogenetic), from those that are artifacts from viral overexpression through use of proper controls. We (this study) and others [29] have shown that viral overexpression of Cre recombinase in the mesencephalon can be toxic and non-specifically cause massive alterations in proteins that regulate dopamine homeostasis and behaviors, which could incorrectly be attributed to genes targeted for Cre/*lox*P recombination. Our findings show that titering down the amount of AAV_Cre used for microinjections can circumvent these non-specific artifacts while retaining Cre/*loxP*-specific recombination, and they underscore the importance of testing the effects of AAV_Cre injections in cohorts of wild-type (non-floxed) controls.

## 4. Materials and Methods

### 4.1. Animals

All studies were conducted in adult group-housed female and male mice 3–4 months of age at the time of injection. Wild-type (WT) C57BL/6J mice (RRID: IMSR_JAX:000664) were purchased from the Jackson laboratories. ErbB4 Cyt-1^fl/fl^ mice harboring *loxP* sites flanking ErbB4 exon 26 were generated as described [20]. In brief, a vector to target ErbB4 exon 26 in embryonic stem (ES) cells was generated by recombineering using vector pL253 [43]. ES cells derived from C57BL/6J mice were used to generate mutant alleles in a C57BL/6 background directly. Successfully targeted ES cells for *ErbB4* exon 26 were enriched by positive/negative (neomycin/ganciclovir) selection, screened by PCR, and injected into blastocysts from albino C57BL/6J mice (JAX stock # 000058). ErbB4 Cyt-1 null mice that were generated by crossing ErbB4 Cyt-1^fl/fl^ mice to EIIa-Cre mice for germline transmission [20], are discussed but not used for any experiments herein. This study used a total of 40 Cyt-1^fl/fl^ mice and 36 WT mice (details in Appendix A); a total of 7 mice were excluded from consideration because post-hoc analysis indicated the injection site was incorrect (see below). Mice were kept on a 12–12 h light–dark schedule with access to food and water *ad libitum*. Animals were handled in accordance with the NIH Animal Welfare guidelines and all animal procedures were approved by the NIH Animal Care and Use Committee (ASP# 18-074). This study was not pre-registered. The study design is shown in the time-line diagram in Appendix A.

### 4.2. Stereotaxic Injections and Post-hoc Confirmation of Injection Sites

AAV_Cre (AAV1_Cre-GFP and AAV9_Cre) or control AAV-GFP (AAV1_GFP and AAV9_GFP; see Table 1 and Appendix A for details) were microinjected unilaterally (Figure 4) or bilaterally (Figure 1, Figure 2, Figure 3 and Figure 5) into the ventral tegmental area (VTA) of adult ErbB4 Cyt-1^fl/fl^ mice and WT mice under isoflurane (Baxter) anesthesia (5% induction, 2% maintenance at 2 L/min oxygen or air flow) using a Hamilton syringe (10 μL Neurosyringe, ID 0.48 mm, Hamilton; Figure 1, Figure 2 and Figure 3) or glass capillaries (Nanoject III, Drummond Scientific; Figure 4) as described previously [9,20]. Randomization was performed to assign animals to a group; groups were balanced for sex and within cages when applicable. For injections with the Hamilton syringe, 250 nL of AAV per hemisphere was injected at 100 nL/min using the following stereotaxic coordinates: anteroposterior (AP) −2.8 mm, lateral (L) 0.75 mm (with respect to bregma), ventral (V) −4.7 mm (with respect to brain surface). We injected 200 nL AAV per hemisphere with glass capillaries at a rate of 5 nL/s using the following stereotaxic coordinates: AP −2.8 mm, L 0.75 mm (with respect to bregma), V −4.35 and −4.3 mm (with respect to brain surface; 100 nL at each Z level). Microinjectors were left in place for a total of 10 min before withdrawing to avoid retrograde mass flow. Injections for one cohort/experiment were performed over two to three consecutive days. Several mice of the same group (ctrl, Cre) were injected consecutively and the order of injection was altered between days. Injection was performed on a heating pad (37 °C) and mice were allowed to recover for >30 min post-surgery in their home cage on a heating pad. Moreover, mice were treated with analgesia ketoprofen, immediately following surgery as well as two subsequent days to reduce post-procedural suffering. Mice were group-housed, food-supplemented (diet gel) and allowed to recover for two weeks before being assessed in behavioral tests as described below. Subsequently (approximately four weeks after injection), injection sites were determined by immunostaining using anti-GFP, anti-TH and anti-DAT antibodies (see below). Based on this post-hoc analysis, a total of seven mice had to be excluded from further consideration.

### 4.3. Battery of Behavior Tests

Behavioral tests of adult ErbB4 Cyt-1^fl/fl^ and WT male and female mice were conducted during the light cycle (6:00 am to 6:00 pm) starting two weeks after injection, in the following sequence: open field, elevated plus maze, Y-maze, and prepulse inhibition (if applicable). All apparatuses were cleaned with 70% ethanol between trials. Tracking in the open field, elevated plus maze, Y-maze, was achieved with ANY-maze software (RRID: SCR_014289). Male mice were always tested before female mice, and group (ctrl, Cre) testing was interspersed with each other. Experimenters were not blinded to the testing groups.

#### 4.3.1. Open Field

Locomotor activity was tested in an open field assay (white chamber, 50 cm × 50 cm × 30 cm, center defined as 28 cm × 28 cm, 70–80 lux in the perimeter, 80–90 lux in the center). Mice were habituated to the testing environment for 30 min prior to testing. Mice were permitted to freely explore the maze for 30 min. Center time and traveled distance were analyzed in 5 min bins.

#### 4.3.2. Elevated Plus Maze

The elevated plus maze test for anxiety-like behavior was assessed in a plus-shaped white plastic apparatus (30 cm × 5 cm arms) consisting of two closed arms (18 cm-high black walls, 60–70 lux) and two open arms (130–140 lux) which stood 40 cm above the ground. Mice were habituated for 30 min to the room and permitted to explore the maze for 5 min. Time spent in the open and closed arms was analyzed.

#### 4.3.3. Y-Maze

Working memory was tested in a Y-Maze apparatus (30 cm × 18 cm × 9.5 cm three-arm maze with opaque tan walls, 50 lux). Mice were habituated for 30 min to the testing environment and then permitted to explore the maze for 5 min. Novel arm entries (spontaneous alternation) were identified as entries into the third arm different from the current and previously explored arm. Consecutive arm entry is considered as a single entry but accounted for total arm entries [44]. Percentage of alternation was calculated as [(number of alternations)/(total arm entries−2)] × 100. Mice with less than 10 total arm entries were excluded from analyses.

#### 4.3.4. Prepulse Inhibition (PPI)

Startle response and prepulse inhibition testing was conducted using a standard startle response system (SR-LAB). Mice were acclimated to a 65 dB background noise in the Plexiglas tube of the testing chamber for 5 min on three consecutive days. Startle response in arbitrary units was determined by pseudo-random presentations of tones ranging from 70 to 120 db, in 5 db increments (five tones each) and normalized to the average of 120 db pulses of the cohort. Startle response was not different between groups in all experiments. During PPI testing on the following day, the animals were presented with a pseudorandom sequence of 20 ms prepulse tones (PP; 66, 70, 74, 78 dB; 12 times each) and 40 ms 120 dB pulse pairings with variable inter-trial interval between 10 to 45 s. A 120 db ‘no prepulse’ (NPP) presented 28 times was used to calculate the percentage of prepulse inhibition of startle response as [(average startle to NPP—average startle to PP)/average startle to NPP] × 100.

### 4.4. Immunofluorescence Histochemistry (IHC)

Immunostaining of 50 µm-thick free-floating sections of AAV-injected mice was performed as previously described [45]. Briefly, mice were anaesthetized with 2.5% avertin and transcardically perfused with 4% paraformaldehyde (PFA) in 0.1 M PBS, pH 7.4, after the completion of behavioral testing, approximately four weeks after injection. Dissected brains were post-fixed overnight in the same fixative at 4 °C and 50 µm-thick sequential sections cut on a vibratome. Sections were blocked in 10% normal donkey serum, 0.3% Triton X-100 (Thermo Fisher, Waltham, MA, USA, Cat No. 28314) in 0.1 M PBS for 1 h at RT and incubated with primary antibodies in blocking solution overnight at 4 °C. Following three 10 min washes in 0.1 M PBS with 0.25% Triton X-100, secondary antibodies were incubated in blocking buffer for 2 h at RT. Samples were extensively washed with 0.1 M PBS, counterstained with DAPI and mounted with Mowiol-DABCO. Primary antibodies used were chicken anti-GFP (1:2000; Thermo Fisher, Waltham, MA, USA, Cat No. A10262, year: 2015; RRID: AB_2534023), rat monoclonal DAT (1:200; clone 6-5G10, Santa Cruz, Dallas, TX, USA, Cat No. sc-32258, year: 2018; RRID: AB_627400) and rabbit TH (1:3000; Pel Freez, Rogers, AR, USA, Cat No. P40101-150, year: 2019; RRID: AB_2617184). Signal was visualized with fluorophore-conjugated donkey and goat secondary antibodies (all 1:1000; donkey anti-rat-Cy3, Jackson Immuno Research, West Grove, PA, USA Cat No. 711-165-152, year: 2016; RRID: AB_2307443; donkey anti-chicken-DL488, Jackson Immuno Research, Cat No. 703-485-155, year: 2011; RRID: AB_2340375; goat anti-rabbit-Alexa647, Invitrogen, Waltham, MA, USA, Cat No. A212245, year: 2019; RRID: AB_2535813). Sections were imaged at 10× magnification using a confocal fluorescent microscope (Zeiss LSM800, Oberkochen, Germany). Average TH and DAT intensity were analyzed using Fiji/ImageJ (RRID: SCR_002285). Area analyzed did not differ between groups in all experiments. Intensity of different cohorts were normalized within each cohort. Mis-injected mice were excluded from all analyses.

### 4.5. Experimental Design and Statistical Analyses

Statistical analyses were performed using Graph Pad Prism 8 (RRID: SCR_002798) and JASP (Version 0.11.1; RRID: SCR_015823). All data represent the mean ± SEM and statistical significance was set at *p* < 0.05. Outliers (ROUT, Q = 1%) were excluded from data analyses. Experimental sample size was set to 5–11 mice/group based on previous behavioral experiments in our laboratory [9]; a statistical method was not used. All data were examined for normality using Shapiro–Wilk test and for equality of variance using Levene’s test or Mauchly’s sphericity test. Statistical analyses were performed using repeated measures two-way ANOVA and Sidak’s multiple comparisons test (Open Field, Elevated Plus Maze, Prepulse inhibition), and unpaired *t* test (Y-maze, TH & DAT intensity). If data points were missing (due to outlier exclusion), a mixed-effect analysis was performed. For data with different variances, Geisser–Greenhouse or Welch’s correction was applied.

## Figures and Tables

**Figure 1 ijms-23-09462-f001:**
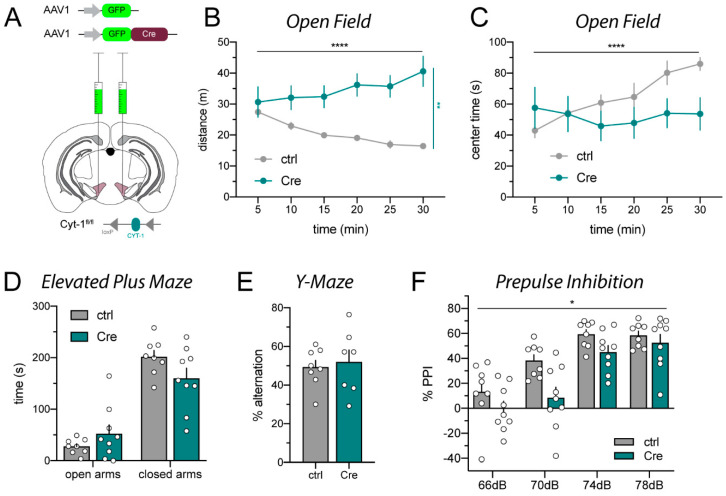
AAV1_Cre-GFP injections into the VTA of ErbB4 Cyt-1^fl/fl^ mice cause hyperactivity and sensorimotor gating deficits relative to AAV1_GFP injected ErbB4 Cyt-1^fl/fl^ controls. (**A**) Scheme visualizing bilateral injection of Cre (AAV1_Cre-GFP) and control (AAV1_GFP; both 10^13^ GC/mL; for details see Appendix A) into the VTA of adult ErbB4 Cyt-1^fl/fl^ mice. (**B**,**C**) Locomotor activity in a novel environment was analyzed in the open field (ctrl n = 8, Cre n = 9). Cre-injected mice became hyperactive compared to control-injected mice ((**B**), two-way ANOVA, F(5,75) = 6.871, *p* < 0.0001 ****; genotype: *p* = 0.0013 **) and tend to spend less time in the center over time ((**C**), two-way ANOVA, F(5,75) = 7.106, *p* < 0.0001 ****; genotype: *p* = 0.2769). (**D**) Anxiety assessed in the elevated plus maze is unchanged (two-way ANOVA, F(1,15) = 2.848, *p* = 0.1122; genotype: *p* = 0.3752; ctrl n = 8, Cre n = 9). (**E**) Spontaneous alternation in the Y-maze is unaltered (unpaired *t* test, *p* = 0.7095; ctrl n = 8, Cre n = 7). (**F**) Prepulse inhibition is impaired in Cre-injected mice compared to controls (two-way ANOVA, F(3,45) = 2.237, genotype: *p* = 0.0334 *; ctrl n = 8, Cre n = 9).

**Figure 2 ijms-23-09462-f002:**
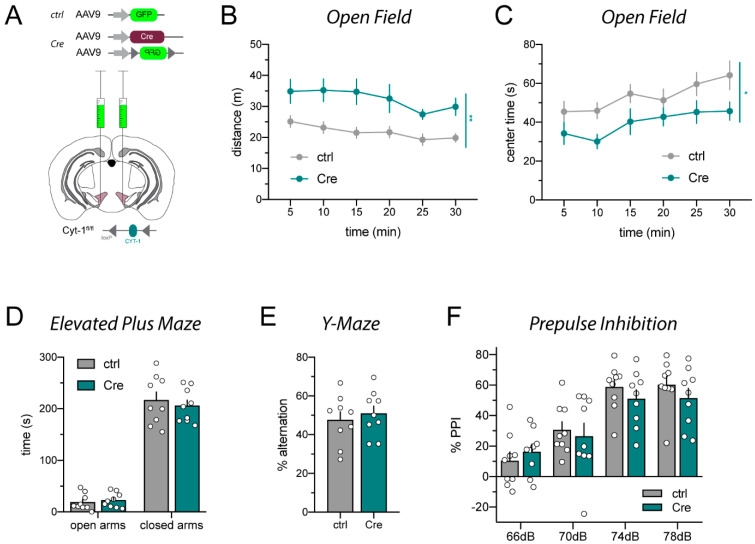
AAV9_Cre injections into the VTA of ErbB4 Cyt-1^fl/fl^ mice cause hyperactivity relative to AAV9-injected ErbB4 Cyt-1^fl/fl^ controls expressing GFP. (**A**) Scheme visualizing bilateral injection of AAV9_Cre/AAV9_DIO-GFP (Cre) and control AAV9_GFP (both 10^13^ GC/mL, for details see Appendix A) into the VTA of adult ErbB4 Cyt-1^fl/fl^ mice. (**B**) AAV9_Cre-injected mice become hyperactive compared to control-injected mice in the open field (ctrl n = 9, Cre n = 9; mixed-effects model, F(5,77) = 0.6211, *p* = 0.6841 and genotype: *p* = 0.0083 **) and spent less time in the center of the maze ((**C**); ctrl n = 9, Cre n = 9; two-way ANOVA, F(5,80) = 0.4144, *p* = 0.8374 and genotype: *p* = 0.0327 *). (**D**) Anxiety measured in the elevated plus maze (ctrl n = 9, Cre n = 9; two-way ANOVA, F(1,16) = 0.3410, *p* = 0.5674 and genotype: *p* = 0.6416). (**E**) Spontaneous alternation in the Y-maze (ctrl n = 9, Cre n = 9; unpaired *t*-test, *p* = 0.5718). (**F**) Prepulse inhibition did not differ between groups (ctrl n = 9, Cre n = 9; two-way ANOVA, F(3,48) = 1.431, genotype: *p* = 0.6086).

**Figure 3 ijms-23-09462-f003:**
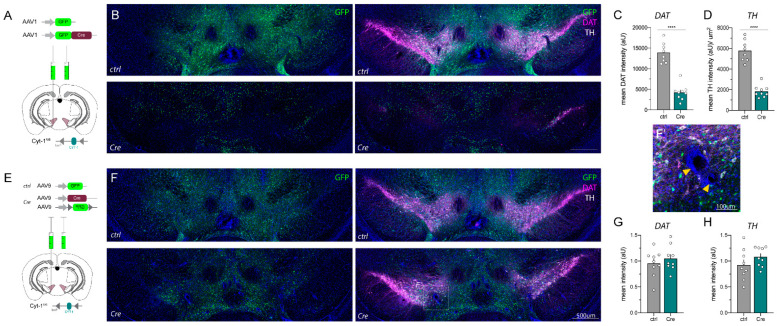
Loss of key dopamine regulatory proteins and appearance of lesions after AAV_Cre injections into the VTA of ErbB4 Cyt-1^fl/fl^ mice. (**A**,**E**) Schematic depiction of microinjections of AAV1 and AAV9 serotypes driving expression of either Cre-recombinase, GFP-tagged Cre or GFP controls (all at 10^13^ GC/mL) into the VTA of adult ErbB4 Cyt-1^fl/fl^ mice. (**B**) AAV1_Cre-GFP injection cause a dramatic loss of DAT (magenta) and TH (white) protein in the mesencephalon after four weeks, quantified in (**C**,**D**) as mean intensity (ctrl n = 8, Cre n = 9; unpaired *t*-test, *p* < 0.0001 ****). (**F**) Immunohistochemical visualization of AAV9_Cre/AAV9_DIO-GFP injections show unaltered protein levels. (**F’**) Magnification of DAPI-negative holes in AAV9_Cre injected mice. (**G**,**H**; ctrl n = 9, Cre n = 9; unpaired *t*-test, DAT *p* = 0.4419, TH *p* = 0.1922), but cause nuclei (DAPI)-negative lesions in the mesencephalon. Scale bars 500 µm in (**B**,**F**) and 100 µm in (**F’**).

**Figure 4 ijms-23-09462-f004:**
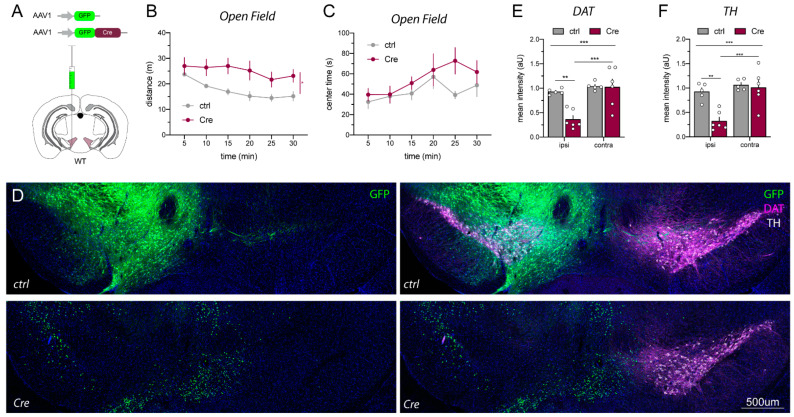
DAT and TH loss in the VTA of AAV1_Cre-GFP-injected WT mice coincide with hyperactivity. (**A**) Scheme visualizing unilateral injection of Cre (AAV1_Cre-GFP) and AAV1_GFP controls at 10^13^ GC/mL into the VTA of adult WT C57BL/6J mice (for details see Appendix A). (**B**,**C**) Cre-injected WT mice become hyperactive compared to GFP control-injected mice in the open field (ctrl n = 5, Cre n = 6; mixed-effect analysis, distance: F(2.327,20.47) = 8.179, *p* = 0.4653 and genotype: *p* = 0.0465 *, center time: F(5,45) = 1.380, *p* = 0.2498 and genotype: *p* = 0.2969). (**D**) Immunostainings of unilaterally injected WT mice with either AAV1_GFP (ctrl; top panels) and AAV1_Cre-GFP (Cre; bottom panels) four weeks after injection. (**E**,**F**) Quantification revealed a massive loss of TH (white) and DAT (magenta) in the ipsilateral (ipsi, injected) side of Cre-injected mice compared to the contralateral (contra, non-injected) side and AAV1_GFP control-injected mice (ctrl n = 5, Cre n = 6; two-way ANOVA, DAT: F(1,9) = 12.12, *p* = 0.0006 ***; ipsi ctrl vs. Cre *p* = 0.0019 **; Cre ipsi vs. contra *p* = 0.0003 ***; TH: F(1,9) = 9.171, *p* = 0.0143 ***; ipsi ctrl vs. Cre *p* = 0.0011 **; Cre ipsi vs. contra *p* = 0.0006 ***). Scale bar 500 µm.

**Figure 5 ijms-23-09462-f005:**
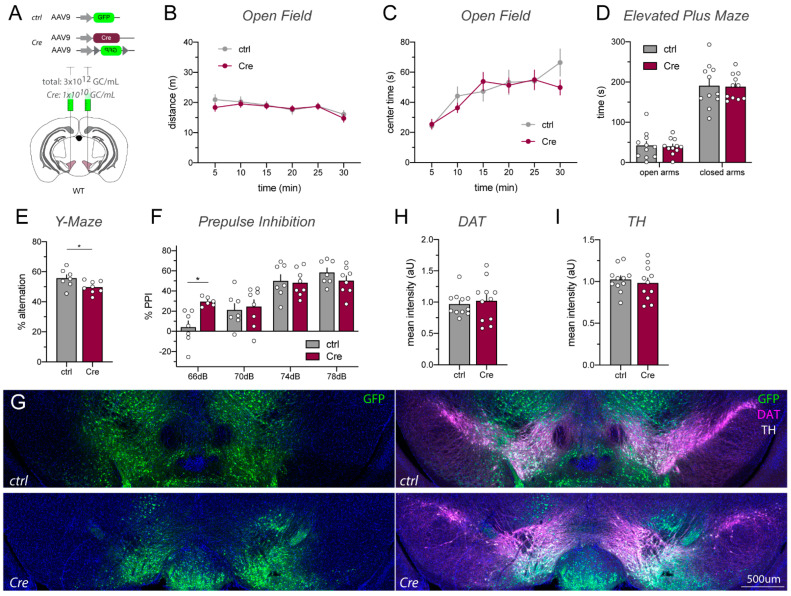
VTA injections of AAV9_Cre at lower titers do not alter behavioral or histological phenotypes. (**A**) Scheme visualizing bilateral injection of AAV9_Cre (10^10^ GC/mL) mixed with AAV9_DIO_GFP (2.99 × 10^12^ GC/mL; mixture hereafter simply denoted as Cre) and control AAV9_GFP (3.0 × 10^12^ GC/mL) into the VTA of adult WT C57BL/6J mice. (**B**,**C**) Distance traveled and center time spent in the open field task does not differ between Cre- and AAV9_GFP control-injected mice (n = 11/group; (**B**) two-way ANOVA, F(5,100) = 0.7684, *p* = 0.5747 and genotype: *p* = 0.5909; (**C**) mixed effects model, F(5,96) = 1.866, *p* = 0.1074 and genotype: *p* = 0.4474). (**D**) Anxiety assessed in the elevated plus maze is unaffected in Cre- and control-injected mice (n = 11/group; two-way ANOVA, F(1,20) = 0.00036, *p* = 0.9851 and genotype: *p* = 0.6528). (**E**) Spontaneous alternation measured in a Y-maze is slightly decreased in Cre- compared to control-injected mice (ctrl n = 7, Cre n = 8; unpaired *t*-test, *p* = 0.0442 *). (**F**) Sensorimotor gating is largely unaffected (ctrl n = 7, Cre n = 8; mixed effects model, F(2.184, 26.93) = 43.11, genotype: *p* = 0.5549, 66dB *p* = 0.0243 * (exclusion of two Cre values (~−30%) at 66 dB in outlier test, otherwise data would be non-significant). (**G**) Immunohistochemistry revealed normal histology in Cre-injected mice. (**H**,**I**) DAT and TH expression are comparable between Cre- and control-injected mice. Scale bars 500 µm.

**Table 1 ijms-23-09462-t001:** AAV constructs used in this study. If applicable, virus titer was adjusted in sterile PBS.

Virus	Catalog # (Addgene)	Plasmid RRID	Concentration (GC/mL)	Figure
AAV1.hSyn.HI.eGFP-Cre.WRPE.SV40	105540	Addgene_105540	2.6 × 10^13^	Figure 1, Figure 3 and Figure 4
AAV1.hSyn.eGFP.WRPE.bGH	105539	Addgene_105539	2.7 × 10^13^	Figure 1, Figure 3 and Figure 4
AAV9.hSyn.Cre.hGH *mixed with* AAV9.synP.DIO.EGFP.WRPE.hGH	105555 100043	Addgene_105555 Addgene_100043	1.5 × 10^13^ 1.5 × 10^13^	Figure 2 and Figure 3
AAV9.hSyn.eGFP.WRPE.bGH	105539	Addgene_105539	3.0 × 10^13^	Figure 2 and Figure 3
AAV9.hSyn.Cre.hGH *mixed with* AAV9.synP.DIO.EGFP.WRPE.hGH	105555 100043	Addgene_105555 Addgene_100043	1.0 × 10^10^ 2.99 × 10^12^	Figure 5
AAV9.hSyn.eGFP.WRPE.bGH	105539	Addgene_105539	3.0 × 10^12^	Figure 5

## Data Availability

Data not contained within this article or Appendix A is available upon request.

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
