# Peer review of "Toxic and Phenotypic Effects of AAV_Cre Used to Transduce Mesencephalic Dopaminergic Neurons"

_ijms, 2022, doi:10.3390/ijms23169462_

Round 1
Reviewer 1 Report
The manuscript from Erben et al. describes the toxic effect on stereotaxic injection of AAV1_Cre in the brain at high titers, and it can cause reductions in gene expression and neurotoxicity in the Mesencephalic Dopaminergic Neurons, leading to behavioral phenotype. The study also highlights the importance of using the proper control while injecting AAV1_Cre recombinase in any targeted gene-floxed mice, which is lacking in many studies. The study is well designed and relevant for targeted gene downregulation studies. However, I have a few concerns that are mentioned below:
1. The authors have used AAV1-GFP as a control for AAV1_Cre-GFP. It might be possible that AAV1-GFP also has a toxic effect on Mesencephalic Dopaminergic Neurons, which could be less than AAV1-Cre. Whether the authors have used any WT mice without any AAV1 injection to compare the difference in any parameters between WT mice (without any AAV1 injection) and AAV1-GFP?
2. Whether the authors have injected AAV1-Cre-GFP in any other neurons apart from Mesencephalic Dopaminergic Neurons to see how unspecific the toxicity of Cre is?
3. It is an interesting finding that reducing the titer of AAV1_Cre-GFP by 1000-fold (10^10 GC/mL) will reduce the unspecific toxic effects of Cre. So, on what basis did the authors decide to reduce the dose by 1000-fold? It would be ideal to show the data on titer-dependent neurotoxicity.
4. English needs to be improved, and a few words that are consequently repeated can be taken care of by the authors.
Reviewer 2 Report
The authors of the present manuscript describe the toxic effects of Cre-recombinase gene-harboring AVV vectors applied for the transduction of mesencephalic dopaminergic neurons. The authors describe behavioral and histological outcome of AAV-Cre vectors infections to the VTA region in the brain, independent of loxP sites recombination. The authors conclusion on the dosing of AAV-Cre vectors and using appropriate controls are valid and highly important for studies applying similar paradigm.
Below are some comments on the proposed manuscript:
1) The authors conclusion in section 4.3 of the discussion claiming that: “Optimizing the concentration of the virus by sequential dilutions to 10^10 GC/mL…prevented reductions in TH/DAT levels…” is misleading. The authors applied concentration dilution only to AAV9 serotype, which did not cause any reduction in the levels TH/DAT even at a higher titer. Thus, we don’t have evidence that reducing AAV titer would prevent reduction in TH/DAT levels, as observed by AAV1-Cre serotype.
2) The differences in behavioral and neurotoxicity outcome between the different serotypes, may very well be the results of different transduction efficiencies… This possibility could have been resolved by comparing the expression levels of Cre-recombinase in AVV1-Cre and AAV9-Cre transduced mice. This should have been done shortly after transduction as in the long-term transduced cells probably die. Nevertheless, it should be suggested for future studies.
3) The figures do not appear in their correct place in the manuscript. All appear in the methods section.
4) In the second paragraph of the introduction the words “in the” are repeated twice in the third sentence.
Round 2
Reviewer 1 Report
The authors have addressed all the comments, this manuscript is acceptable now.